# Patient-Reported Functional Outcomes and Quality of Life After Contact X-Ray Brachytherapy (CXB) in Organ-Preserving Management of Rectal Cancer

**DOI:** 10.3390/cancers17091560

**Published:** 2025-05-03

**Authors:** Ngu Wah Than, D. Mark Pritchard, David M. Hughes, Carrie A. Duckworth, Muneeb Ul Haq, Thomas Cummings, Charlotte Jardine, Sarah Stead, Rajaram Sripadam, Arthur Sun Myint

**Affiliations:** 1Department of Molecular and Clinical Cancer Medicine, Institute of Systems, Molecular and Integrative Biology, The University of Liverpool, Liverpool L69 3GE, UK; nwthan@liverpool.ac.uk (N.W.T.); carried@liverpool.ac.uk (C.A.D.); muneeb.haq@liverpool.ac.uk (M.U.H.); 2The Clatterbridge Cancer Centre NHS Foundation Trust, 65 Pembroke Place, Liverpool L7 8YA, UK; thomas.cummings@nhs.net (T.C.); charlotte.jardine@nhs.net (C.J.); sarah.stead1@nhs.net (S.S.); rajaram.sripadam@nhs.net (R.S.); 3Department of Health Data Science, Institute of Population Health, The University of Liverpool, Liverpool L7 3EA, UK; dmhughes@liverpool.ac.uk

**Keywords:** rectal cancer, contact X-ray brachytherapy, Papillon, radiotherapy, organ preservation, patient-reported outcomes, functional outcomes, health-related quality of life

## Abstract

Contact X-ray brachytherapy (CXB) in combination with (chemo)radiation is used as an organ-preserving treatment in rectal cancer management. However, data on health-related quality of life (HRQOL) outcomes after CXB remain limited. This prospective observational study assessed HRQOL after CXB with a one-year follow-up. CXB combined with (chemo)radiation maintained stable HRQOL with some improvements. These findings suggest that CXB treatment, when combined with (chemo)radiation, does not compromise HRQOL of patients and helps them consider the pros and cons of CXB as a treatment option.

## 1. Introduction

Neoadjuvant treatments such as total neoadjuvant therapy, (chemo)radiotherapy, and immunotherapy for dMMR/MSI-H (mismatch repair deficient/microsatellite instability high) tumours, followed by radical total mesorectal excision (TME), have become the standard approach for rectal cancer management [1,2,3]. This multi-modal strategy significantly enhances tumour control and offers promising survival benefits [4,5]. With advancements in rectal cancer management leading to longer patient survival, the impact of treatment on the quality of life (QOL) of patients has become an important focus of attention.

TME surgery remains associated with postoperative complications due to multiple factors [6] and long-term impairment of QOL. There are also some challenges related to the surgical approach, particularly when accessing lesions through the anus, especially for semicircular or anterior rectal wall lesions, which can potentially lead to nerve damage, resulting in sexual-related complications. These effects are more pronounced when combined with neoadjuvant treatment. Bowel dysfunction, especially low anterior resection syndrome (LARS), is the most common and severe consequence. It presents with a range of symptoms beyond faecal incontinence, including frequent bowel movements, urgency, and clustering, with some effects persisting for up to 14 years [7,8,9,10]. Furthermore, urinary and sexual functions, and emotional and psychosocial well-being are also significantly affected in both sexes [11,12,13].

For over two decades, studies on organ-preserving strategies including contact X-ray brachytherapy (CXB) boost combined with (chemo)radiation, have demonstrated their effectiveness in achieving long-term organ preservation, disease-free survival, and overall survival [14,15,16,17]. These non-surgical approaches are increasingly favoured by both patients and clinicians, as they not only reduce surgery-related complications and stoma formation but also enhance long-term oncological outcomes and potentially improve patients’ QOL [18,19].

Multiple functional outcome studies have examined anorectal function, as well as health-related quality of life (HRQOL) in bowel, urinary, and sexual functions, in the contexts of watch-and-wait after (chemo)radiation, local excision alone, and local excision following (chemo)radiation [20,21,22,23]. However, only one small study to date has assessed functional outcomes in inoperable and older patients after CXB combined with (chemo)radiation [24]. Detailed data on HRQOL, emotional well-being, and general health function after CXB treatment, therefore, remain limited.

Our study aimed to assess patient-reported HRQOL in rectal adenocarcinoma patients who received CXB combined with (chemo)radiation as an organ-preserving treatment option. In order to assess HRQOL in a real-world clinical setting, we included patients who received CXB for a range of different clinical indications. We focus on HRQOL outcomes, as well as emotional and general health status over the course of a year after CXB treatment.

## 2. Materials and Methods

Ethics committee approval was not required since this was a prospective observational study conducted during routine patient care. We obtained institutional audit approval on 3 May 2022. Permission to use the EORTC-QLQ-CR29 was granted by the European Organization for Research and Treatment of Cancer (EORTC), and approval for the EQ-5D-3L questionnaire was obtained from EuroQol.

### 2.1. Patient Selection

All consecutive patients who were referred to the Clatterbridge Cancer Centre (CCC), Liverpool, from January to October 2023, for CXB treatment as an organ-preserving approach, either before or after (chemo)radiation, were eligible to participate. Patients who received CXB as postoperative adjuvant treatment following local rectal cancer excision (in whom a lower dose of 60 Gy is usually employed) were not included as these patients do not usually have any tumour-related symptoms. Patients who underwent CXB for palliative purposes/symptom control alone were also excluded, as these patients were mostly frail and were, therefore, unlikely to be able to undergo the regular follow-up required for the study. Eligible patients were provided with a patient information sheet and invited to participate in the study (Appendix A). Verbal informed consent was subsequently obtained at every study time point. The total number of consecutive patients and the selection pathway are demonstrated in Figure 1A.

### 2.2. Data Collection

Data were collected with a one-year follow-up from the start of CXB treatment. The study time points included baseline (at the start of CXB treatment), at the end of CXB treatment, 6 months from baseline, and 12 months from baseline. Baseline and end-of-treatment data were collected in the treatment waiting room using physical forms, while 6-month and 12-month follow-ups were conducted via telephone appointments using electronic forms. All data collection was carried out by a single dedicated staff member (NWT).

### 2.3. Questionnaires

Our study utilised three sets of questionnaires: the EORTC-QLQ-CR29, a colorectal cancer-specific questionnaire validated by the EORTC; the Hospital Anxiety and Depression Scale (HADS) to assess emotional well-being; and the EuroQol EQ-5D-3L to evaluate patients’ general health status.

The EORTC-QLQ-CR29 consists of 29 items, including 12 related to gastrointestinal and urinary symptoms, 6 addressing mental health, 7 separate questions for anorectal function tailored for participants with or without a stoma, and 4 specific questions assessing sexual function in men and women. The HADS comprises 14 questions, with 7 assessing anxiety and 7 evaluating depression. The EQ-5D-3L includes an index score based on five dimensions: mobility, self-care, usual activities, pain/discomfort, and anxiety/depression, each of which has three levels of response (no problems, some problems, extreme problems/unable to), along with a visual analogue scale (EQ-VAS) with 0–100 scales representing the patient’s self-rated health.

### 2.4. Statistical Analysis

The EORTC-QLQ-CR29 questionnaires were processed following the EORTC scoring manuals, with scores ranging from 0 to 100, where higher scores indicate better functioning and global health or a greater level of symptoms [25]. Overall scores for each scale were calculated as group means with standard deviations (mean ± SD), as is commonly reported in most QOL studies, and analysed across time points. Sexual symptom scores for men (impotency) and women (dyspareunia) were combined for analysis due to the low completion rates for these scores. Each HADS score ranged from 0 to 3, with overall scores classified as follows: 0–7 (normal), 8–10 (borderline abnormal or borderline case), and 11–21 (abnormal or case—indicating the need for referral to a specialised mental health assessment). Data were presented as counts and percentages across different time points. The EQ-5D-3L questionnaires were scored according to their scoring manuals [26] and reported as group means with standard deviations (mean ± SD). Missing data were handled by the pairwise deletion method, and the remaining dataset was analysed [27].

To assess clinically meaningful changes in quality of life (QOL), we calculated the standardized effect size (ES) using the formula (Cohen’s d = (mean2 − mean1)/pooled standard deviation) according to Cohen [28]. ES values were interpreted as follows: “no change” (ES < 0.2), “small change” (ES = 0.2–0.4), “moderate change” (ES = 0.5–0.7), and “considerable change” (ES ≥ 0.8). In addition, we reported the proportion of patients who experienced a clinically relevant deterioration in domain scores on the HAD scales, defined as a decrease/increase of more than 10 points from baseline (i.e., 10% of the scale range), as recommended by Osoba et al. [29].

Longitudinal analyses were subsequently conducted using a linear mixed-effects (LME) model to account for inherent biases in treatment and disease characteristics among patients, and to capture the correlation between repeated measurements on the same individual [30]. Each domain score was fit with separate LME models to assess changes in HRQOL scores over time, incorporating fixed effects (time effect, treatment intent, radiation regimen, and residual/regrowth disease status at specific study time points, and multivariate) and random effects (repeated measures within subjects) [31]. All statistical analyses were performed using R version 4.4.0.

## 3. Results

A total of 53 patients initially agreed to participate in the study at baseline, with 51 remaining at the end of CXB treatment, 47 at the 6-month follow-up, and 42 at the 12-month follow-up. The study flowchart illustrating participant retention over time is shown in Figure 1B. Almost all patients completed all subsets of questionnaires (100% at baseline, 96% at end of treatment, 89% at 6 months, 79% at 12 months) with some missing sexual symptoms and function scores subsets (26–62% of completion throughout follow-up period) due to the relatively older age and sexually inactive nature of some members of the study population.

The median age of patients was 71 years (IQR: 64.0–77.5), with a male predominance (64%) and an initial TNM stage of 1–3. Among the participants, 49% received CXB for residual disease following (chemo)radiation, 23% were de novo patients who underwent CXB first, followed by subsequent (chemo)radiation, and 28% received CXB for local regrowth after a period of ‘watch-and-wait’ following (chemo)radiation alone. In total, 77% of participants had, therefore, completed a course of (chemo)radiation treatment prior to enrolment, while 23% received (chemo)radiation during the follow-up period of this study after completing CXB.

By the end of the study follow-up, 30 (56%) out of 53 patients achieved a clinical complete response, 21 (40%) had residual disease, and 2 (4%) experienced local regrowth. A total of 8 out of 21 patients who experienced residual disease underwent subsequent surgery, while 3 had palliative colostomy, and all remaining patients received optimal supportive care. Distant relapse occurred in 6 patients (11%), including 4 (7%) who had synchronous residual disease or local regrowth, and 4 of them received palliative chemotherapy. One patient experienced an isolated pelvic nodal relapse and subsequently underwent stereotactic body radiotherapy. Additionally, three patients died at the one-year follow-up; one patient who had achieved a complete response, one who had residual disease, and another with isolated metastatic lung disease. These overall oncological outcomes are poorer than those reported in some previous studies but probably reflect the heterogeneous indications for treatment in this cohort of patients. In particular, the cohort included 28% of patients who had experienced local regrowth after a watch-and-wait approach and in whom poorer long-term outcomes were expected at the time of treatment [32]. Only 5/15 (33%) patients in this subgroup achieved a complete clinical response following CXB. Detailed patient, treatment characteristics, and oncological outcomes are presented in Table 1 and Appendix A. The individual outcomes of each subgroup were not analysed statistically, as the sample size for each was considered too small to allow meaningful interpretation, and the outcomes of each subgroup have already been explored with larger patient cohorts in previous studies [32,33,34].

### 3.1. EORTC-QLQ-CR29 Scores

Generally, patient-reported symptoms and functional scores remained stable throughout the follow-up period, without evidence of worsening of symptoms from baseline. The mean scores for bowel symptoms showed no significant changes over the year on LME models, including rectal bleeding, stool frequency, faecal incontinence, bloating, embarrassment, buttock pain, and sore skin, although considerable changes were observed from time to time for stool frequency and bloating scores. However, significant improvements were observed in abdominal pain (mean ± SD score: 5.03 ± 2.08 vs. 0.79 ± 0.79; coefficient (from linear mixed models): −1.85; 95% CI: −3.49, −0.20, *p* = 0.03) and flatulence (mean ± SD score: 36.48 ± 4.51 vs. 22.22 ± 3.71; coefficient: −5.93; 95% CI: −9.16, −2.71, *p* < 0.001) at 12 months. Additionally, urinary frequency (mean± SD score: 27.67 ± 3.48 vs. 16.67 ± 3.64; coefficient: −4.63; 95% CI: −6.90, −2.37, *p* < 0.009) statistically improved over the year, while dysuria and urinary incontinence scores remained unchanged.

Regarding functional scales, anxiety scores remained insignificant in the LME model despite a considerable improvement being observed over time. However, the body image scores were significantly influenced by disease status and subsequent salvage surgery at the time of follow-up in multivariate analysis. Additionally, there was a significant improvement in body weight after completing treatment on the LME model (mean ± SD: 85.53 ± 24.90 vs. 92.06 ± 16.15; coefficient: 3.35; 95% CI: 0.82–5.86; *p* = 0.01).

Sexual symptoms (impotence/dyspareunia) and functional (interest) scores showed no significant changes in the LME models over the year. However, these results may not fully represent the true outcomes due to the low completion rates of these assessments. Figure 2 illustrates the results of all symptom and functional scores.

### 3.2. HADS Scores

Most patients maintained emotional stability throughout the follow-up period, with a slight improvement in normal anxiety scores (83–88%) and a minor decrease in the percentage of those with normal depression scores (94–88%), indicating no significant deterioration over time (Figure 3).

### 3.3. EQ-5D-3L Scores

The mean ±SD scores of the EQ-5D health index remained between 0.85 ± 0.19 and 0.82 ± 0.16 while EQ-VAS scores showed a statistically significant improvement from baseline to 12 months in the LME model (mean ± SD: 70.38 ± 17.15 vs. 77.86 ± 14.78; coefficient: 1.84; 95% CI: 0.43, 3.26, *p*= 0.01], indicating a good overall quality of life after CXB treatment (Figure 4). Details of the score results for all questionnaires and LME analyses are also provided in Table 2 and Appendix A.

## 4. Discussion

Various studies [20,21,22] have reported different forms of HRQOL data in rectal cancer patients following (chemo)radiation as an organ-preserving strategy. Our study adds to this by providing information on functional outcomes and quality of life in patients in a real-world setting who received CXB in addition to neoadjuvant (chemo)radiation as part of an organ-preserving approach.

Overall, patient-reported symptoms and functional scores in this study showed minimal variation throughout the follow-up period, with no apparent worsening of symptoms or functions at one year. Bowel symptoms, particularly faecal incontinence, frequency, rectal bleeding, and embarrassment remained stable from their initial status, while significant improvements were observed in abdominal pain and flatulence. These findings were largely consistent with the CXB outcomes reported by Custers et al. [24] and with (chemo)radiation alone, as reported by Dizdarevic et al. [21], except for rectal bleeding, which worsened up to 24 months in that study. By contrast, the major LAR scores were noted in nearly a quarter of patients for up to two years after (chemo)radiation alone or in combination with local excision, as reported by Custers et al. [22] and Jones et al. [23].

Although higher levels of urinary dysfunction are typically reported following neoadjuvant radiotherapy [11,35], our findings suggested an improvement in urinary frequency. This could be explained by the fact that urinary frequency, which was initially caused by bladder irritation from the large tumour, was alleviated once the tumour had shrunk and the symptoms following (chemo)radiation had already resolved over time.

Radiotherapy has also been linked to a decline in sexual function in previous reports [11,36]. However, our study’s findings were unable to fully capture this aspect of QOL, because of the older age and sexually inactive status of several members of the study population.

The gradual improvement in anxiety scores on the EORTC and the HADS anxiety scales reflect a decrease in patients’ concerns about tumour recurrence and the risk of subsequent surgery over time, contributing to better emotional well-being. This is further supported by the significant improvement in patients’ self-rated health status, as indicated by the EQ-VAS scores. A similar finding was reported in the studies of (chemo)radiation with/without local excision [22,23].

This study has some limitations. Firstly, it was a relatively small study with mixed data regarding the timing of CXB administration (either upfront (chemo)radiation or CXB) and its indications (CXB as a boost for organ-preserving curative treatment or as salvage for local tumour regrowth after watch-and-wait with (chemo)radiation alone). This variability will have influenced the evaluation of pure functional outcomes following CXB. In particular, those patients who received CXB to treat local tumour regrowth after prior (chemo)radiation had poorer outcomes as we have previously reported [32]. Secondly, the follow-up period was relatively short, limiting our ability to assess long-term outcomes. Thirdly, a few patients underwent additional treatments, including surgery, during the follow-up period. Additionally, there was only one small fully published reference study with which to compare our results for validation. Therefore, larger prospective studies with homogeneous study populations and extended follow-up periods are recommended to fully assess functional outcomes following CXB treatment. QOL assessments were performed during the recently published OPERA trial, but these have only been published in abstract form to date [37].

## 5. Conclusions

This study has confirmed that good functional and HRQOL outcomes were maintained one year after CXB treatment combined with (chemo)radiation and that there were significant improvements in some symptoms and functional outcomes. These findings could provide valuable information for counselling patients and facilitate shared decision-making, helping them better understand the pros and cons of CXB as one of the available treatment options in rectal cancer management.

## Figures and Tables

**Figure 1 cancers-17-01560-f001:**
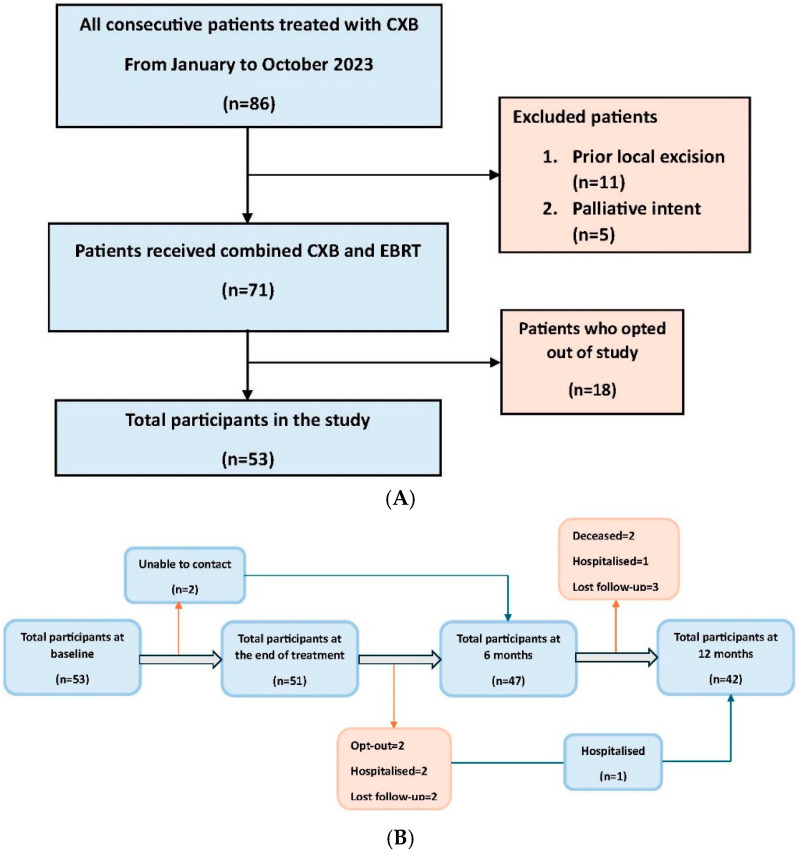
(**A**) Flowchart demonstrating patient eligibility and total participants of the study. (**B**) Flowchart illustrating participant retention over time, with pink arrows representing those who left the study and blue arrows indicating those who returned.

**Figure 2 cancers-17-01560-f002:**
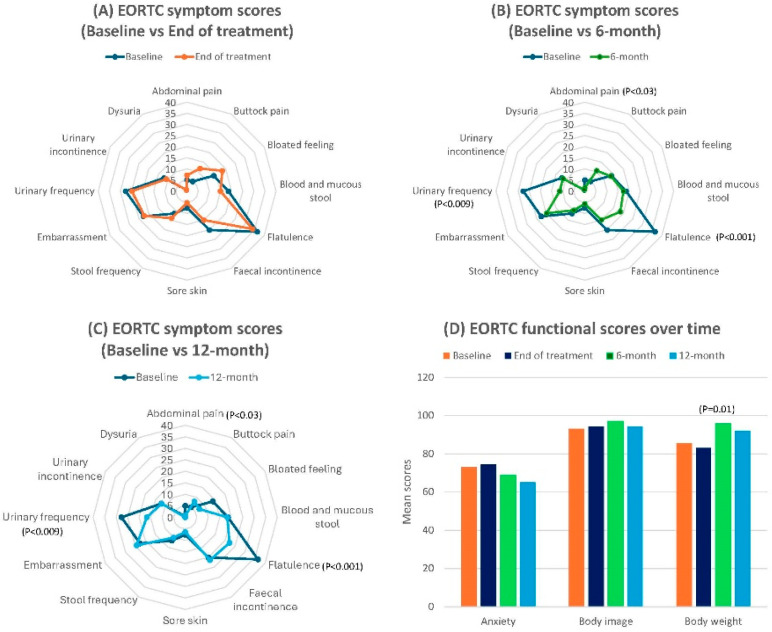
Changes in EORTC symptom mean scores: from baseline to end of treatment (**A**), from baseline to 6 months (**B**), from baseline to 12 months (**C**), and functional mean scores (**D**) over the study period.

**Figure 3 cancers-17-01560-f003:**
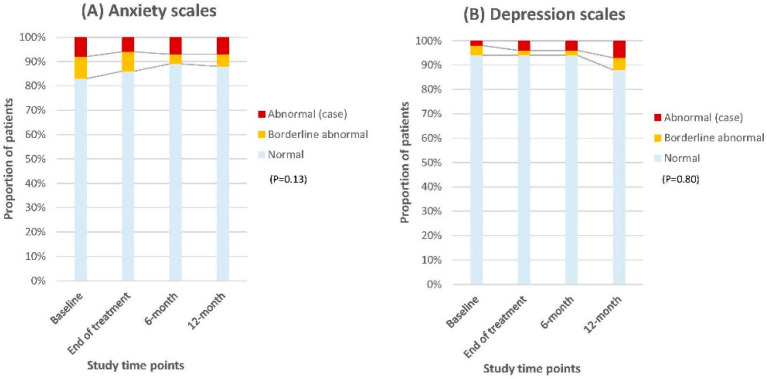
Changes in the proportion of patients with anxiety (**A**) and depression (**B**) scores over time according to Hospital Anxiety and Depression Scales (HADS).

**Figure 4 cancers-17-01560-f004:**
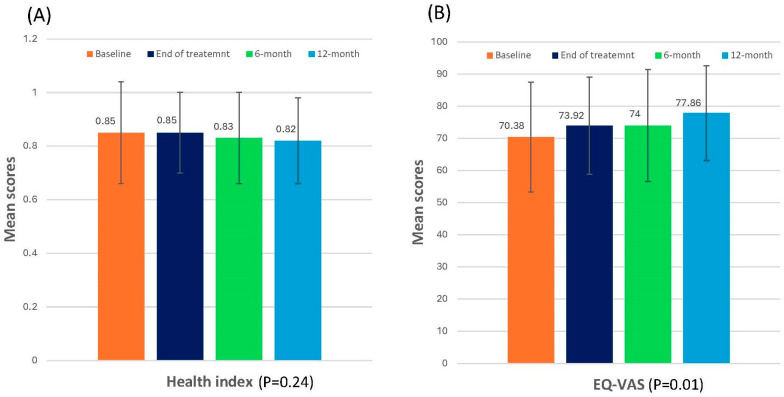
Changes in health index (**A**) and EQ-VAS (**B**). Mean ± standard deviation scores of EQ-5D-3L questionnaire over time.

**Table 1 cancers-17-01560-t001:** Clinicopathological, treatment characteristics, and oncological outcomes.

Characteristics		(Number of Participants = 53) (%)
Median (IQR) Age (Years)		71 (64–77.5)
Sex	Male	34 (64)
	Female	19 (36)
ECOG performance status	0–1	34 (64)
	2–3	19 (36)
Stage	I	16 (30)
	II	15 (28)
	III	19 (36)
	Tx-Nx-Mx	3 (6)
Distance from the anal verge	≤6 cm	38 (72)
	7–10 cm	15 (28)
Treatment intent of CXB	De novo	12 (23)
	Residual	26 (49)
	Regrowth	15 (28)
EBRT regimen	Short course	22 (41)
	Long course	3 (6)
	Chemoradiation	28 (53)
CXB dose	90 Gy	42 (79)
	110 Gy	11 (21)
**Oncological outcomes**	**(Number of participants = 53) (%)**	**Subsequent treatment**
Clinical complete response	30 (56)	
Residual disease	21 (40)	APER = 7 Hartmann’s = 1 Palliative colostomy = 3 Palliative chemotherapy = 2 Supportive care = 8
Local regrowth	2 (4)	Supportive care = 2
Synchronous distant relapse with residual/local regrowth	4 (7)	Palliative chemotherapy = 2 Supportive care = 2
Isolated distant relapse	2 (4)	Palliative chemotherapy = 2
Isolated pelvic node relapse	1 (2)	Stereotactic radiotherapy = 1
Deceased	3 (6)	

**Table 2 cancers-17-01560-t002:** Domain scores of EORTC-QLQ-CR29, EQ-5D-3L, and HADS questionnaires over time.

		EORTC-QLQ-CR29						
Domain Score		Baseline (Mean ± SD)	End of Treatment (Mean ± SD)	Effect Size	6-Month (Mean ± SD)	Effect Size	12-Month (Mean ± SD)	Effect Size
Abdominal pain		5.03 ± 2.08	7.19 ± 2.69	−0.9	1.42 ± 0.99	2.2	0.79 ± 0.79	2.7
Buttock pain		5.03 ± 2.08	11.76 ± 3.34	−2.4	10.64 ± 3.06	−2.1	7.94 ± 2.96	−1.1
Bloated feeling		13.84 ± 3.86	18.30 ± 4.10	−1.1	13.48 ± 2.98	0.1	7.14 ± 2.90	1.9
Blood and mucous stool		18.55 ± 3.23	15.03 ± 2.69	1.2	17.38 ± 2.95	0.4	18.25 ± 3.30	0.09
Flatulence		36.48 ± 4.51	33.99 ± 4.52	0.6	18.44 ± 3.33	4.5	22.22 ± 3.71	3.5
Faecal incontinence		20.13 ± 4.15	15.03 ± 3.53	1.3	14.89 ± 3.01	1.4	21.43 ± 4.22	−0.3
Sore skin		7.55 ± 2.93	5.23 ± 2.16	0.9	5.67 ± 2.11	0.7	6.35 ± 2.04	0.5
Stool frequency		11.64 ± 2.36	14.05 ± 2.21	−1.1	9.93 ± 1.87	0.8	10.16 ± 2.01	0.7
Embarrassment		22.64 ± 4.38	22.22 ± 4.14	0.1	19.86 ± 4.14	0.7	24.39 ± 5.59	−0.4
Urinary frequency		27.67 ± 3.48	24.84 ± 3.6	0.6	11.35 ± 2.33	3.8	16.67 ± 3.64	1.7
Urinary incontinence		11.94 ± 3.37	10.80 ± 2.69	0.4	11.35 ± 2.47	0.2	11.84 ± 2.60	0.03
Dysuria		0.63 ± 0.62	0.65 ± 0.65	−0.03	0.71 ± 0.71	0.1	0.00 ± 0.00	0.1
Sexual symptoms		29.29 ± 6.77	35.63 ± 8.09	−0.9	27.16 ± 7.13	0.3	30.77 ± 8.04	0.2
Sexual interest		22.22 ± 4.95	25.56 ± 6.33	−0.6	21.79 ± 6.12	0.1	15.00 ± 5.12	1.4
Anxiety		72.96 ± 4.12	74.51 ± 3.69	−0.4	68.79 ± 4.80	0.9	65.08 ± 5.07	1.7
Body image		93.08 ± 2.68	94.11 ± 2.84	−0.4	97.00 ± 1.71	−1.7	94.06 ± 2.65	−0.4
Body weight		85.53 ± 24.90	83.00 ± 26.97	0.1	95.74 ± 13.22	−0.5	92.06 ± 16.15	−0.3
		**EQ-5D-3L**						
Score		Baseline (Mean ± SD)	End of treatment (Mean ± SD)	Effect size	6-month (Mean ± SD)	Effect size	12-month (Mean ± SD)	Effect size
Health index		0.85 ± 0.19	0.85 ± 0.15	0.2	0.83 ± 0.17	0.1	0.82 ± 0.16	0.2
EQ-VAS		70.38 ± 17.15	73.92 ± 15.14	−0.2	74.00 ± 17.37	−0.2	77.86 ± 14.78	−0.5
		**HADS**						
Score		Baseline N = 53 (%)	End of treatment N = 51 (%)	Score difference	6-month N = 47 (%)	Score difference	12-month N = 42 (%)	Score difference
Anxiety	Normal	44 (83)	44 (86)	3	42 (89)	6	37 (88)	5
	Borderline	5 (9)	4 (8)	1	2 (4)	5	2 (5)	4
	Abnormal (case)	4 (8)	3 (6)	2	3 (7)	1	3 (7)	1
Depression	Normal	50 (94)	48 (94)	0	44 (94)	0	37 (88)	6
	Borderline	2 (4)	1 (2)	2	1 (2)	2	2 (5)	1
	Abnormal (case)	1 (2)	2 (4)	2	2 (4)	2	3 (7)	5

## Data Availability

Research data are stored in the institutional repository and anonymised data will be shared upon request to the corresponding author.

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
