# Peer review of "Patient-Reported Functional Outcomes and Quality of Life After Contact X-Ray Brachytherapy (CXB) in Organ-Preserving Management of Rectal Cancer"

_cancers, 2025, doi:10.3390/cancers17091560_

Round 1

Reviewer 1 Report

Comments and Suggestions for Authors

Very interesting paper because it deals with an important problem such as the quality of life in patients with rectal cancer who have been directed to the neoadjuvant and subsequently surgical pathway with TME resection which involves organ preservation. Absolutely shareable introduction, in content, even if it is recommended to make a distinction between adenocarcinoma and squamous cell carcinoma. We absolutely agree with the dissertation that colleagues make about TME which is an essential cornerstone in the therapy of non-advanced rectal cancer and that only certain centers have the possibility of being able to deal with due to the need for adequate technology and trained personnel. I would also add something about the type of surgical access, Not simple from the anus especially for lesions that are semicircular or in the anterior wall of the rectum and which can lead together with the boost therapy of X-ray brachytherapy to those that are nerve lesions that result in impotentia erigendi or dyspareunia. Access, which can be considered minimally invasive, in expert hands, is what can mitigate in many cases the effects on continence or sexual activity. Naturally we encourage concentrating patients who present these pathologies in high-volume centers, such as that of the colleagues who wrote the article because this is a precursor to the best results (doi.org/10.3390/jcm12072708 to be read and cited in the bibliography). Measuring the quality of life with coded questionnaires such as the one proposed by colleagues obviously represents a qualified measurement also because it is clear from the writing that it is self-administered without external conditioning by even a simple reader. Excellent bibliography, excellent English, excellent iconography, although we suggest adding images

Author Response

Very interesting paper because it deals with an important problem such as the quality of life in patients with rectal cancer who have been directed to the neoadjuvant and subsequently surgical pathway with TME resection which involves organ preservation.

Thank you very much for your comment.

Absolutely shareable introduction, in content, even if it is recommended to make a distinction between adenocarcinoma and squamous cell carcinoma.

Thank you. This has now been added to Line 80, in the Introduction section (Page 2).

We absolutely agree with the dissertation that colleagues make about TME which is an essential cornerstone in the therapy of non-advanced rectal cancer and that only certain centers have the possibility of being able to deal with due to the need for adequate technology and trained personnel. I would also add something about the type of surgical access, Not simple from the anus especially for lesions that are semicircular or in the anterior wall of the rectum and which can lead together with the boost therapy of X-ray brachytherapy to those that are nerve lesions that result in impotentia erigendi or dyspareunia.

Thank you. We have added as suggested: “There are also some challenges related to the surgical approach, particularly when accessing lesions through the anus, especially for semicircular or anterior rectal wall lesions, which can potentially lead to nerve damage, resulting in sexual related complications.” In the Introduction section. (Line 56-59, Page 2)

Access, which can be considered minimally invasive, in expert hands, is what can mitigate in many cases the effects on continence or sexual activity. Naturally we encourage concentrating patients who present these pathologies in high-volume centers, such as that of the colleagues who wrote the article because this is a precursor to the best results (doi.org/10.3390/jcm12072708 to be read and cited in the bibliography).

Thank you for suggesting the reference to include in the literature review; it has now been cited on Line 56 in the Introduction section (Page 2).

Measuring the quality of life with coded questionnaires such as the one proposed by colleagues obviously represents a qualified measurement also because it is clear from the writing that it is self-administered without external conditioning by even a simple reader. Excellent bibliography, excellent English, excellent iconography, although we suggest adding images.

Thank you for your comment. We have added Supplementary Figure 1, which shows the patient information sheet provided when inviting patients to participate in the study, in the patient selection, Materials and Methods section (Line 101, Page 3).

Reviewer 2 Report

Comments and Suggestions for Authors

The literature review is incomplete.

No MCID is defined for any quality‑of‑life instrument, making subsequent score changes hard to interpret.

Post‑local excision and palliative CXB cases are excluded without scientific justification.

Switching from paper questionnaires at baseline to telephone/electronic collection at follow‑up introduces mode‑of‑administration bias that is not addressed.

Missing‑data handling is not mentioned.

Subgroup heterogeneity (de‑novo vs residual vs regrowth) is acknowledged in the text but never explored statistically.

Author Response

The literature review is incomplete.

No MCID is defined for any quality‑of‑life instrument, making subsequent score changes hard to interpret.

Thank you very much for pointing this out. We have added “To assess clinically meaningful changes in quality of life (QOL), we calculated the standardised effect size (ES) using the formula (Cohen's d = (Mean2 - Mean1) ⁄ Pooled Standard Deviation) according to Cohen [30]. ES values were interpreted as follows: “no change” (ES < 0.2), “small change” (ES = 0.2–0.4), “moderate change” (ES = 0.5–0.7), and “considerable change” (ES ≥ 0.8). In addition, we reported the proportion of patients who experienced a clinically relevant deterioration in domain scores on the HAD scales, defined as a decrease/increase of more than 10 points from baseline (i.e., 10% of the scale range), as recommended by Osoba et al [31].” in the Materials and Methods section (Line 143-150, Page 4).

In addition, we have done additional analysis for this and have added some text in the Results section according to the calculation results (please refer to revised table 2) (Line 201-249, Page 6-9).

Post‑local excision and palliative CXB cases are excluded without scientific justification.-

Thank you. We have added more clarification about the reasons for excluding those patients as: “Patients who received CXB as postoperative adjuvant treatment following local rectal cancer excision (in whom a lower dose of 60Gy is usually employed) were not included as these patients do not usually have any tumour-related symptoms. Patients who underwent CXB for palliative purposes/symptom control alone were also excluded, as these patients were mostly frail and were therefore unlikely to be able to undergo the regular follow-up required for the study.” in the patient selection, Materials and Methods section (Line 97-100, Page 3).  

Switching from paper questionnaires at baseline to telephone/electronic collection at follow‑up introduces mode‑of‑administration bias that is not addressed.

Thank you for the comment. This modification involved transitioning from paper-based to electronic forms, with a single designated staff member (first author) administering the questionnaires face-to-face at the baseline and post-treatment clinic visits and conducting follow-up assessments via telephone throughout the study period. We do not think that this will have biased the results significantly as the same researcher was involved in performing all the assessments.

Missing‑data handling is not mentioned.

Thank you for pointing this out. We handled missing data using the pairwise deletion (simple omission) method and analysed the remaining dataset. This is a widely accepted method for this type of study and has now been mentioned in the Materials and Methods section as “Missing data was handled by the pairwise deletion method, and the remaining dataset was analysed.” (Line 140-141, Page 4).

Subgroup heterogeneity (de‑novo vs residual vs regrowth) is acknowledged in the text but never explored statistically.

Thank you for this comment. We did not perform a statistical analysis of the oncological outcomes for these subgroups despite demonstrating the individual outcomes as a descriptive table (Supplementary table 1), as sample sizes were considered too small to yield statistically meaningful results. However, the outcomes of these sub-groups have been explored with larger numbers of patients in our previous studies (for de-novo cases: 10.1016/j.radonc.2024.110465, residual disease: 10.1016/j.ijrobp.2024.11.113, and regrowth cases: 10.5114/jcb.2024.139049). We have also added the following explanation to the text as “The individual outcomes of each subgroup were not analysed statistically, as the sample size for each was considered too small to allow meaningful interpretation and the outcomes of each subgroup have already been explored with larger patient cohorts in previous studies.” in the Results section (Line 196-199, Page 5).

Reviewer 3 Report

Comments and Suggestions for Authors

Title - clearly depicting the purpose of the study - No remarks

Abstract - a concise review of the manuscript, emphasizing all the important points. 

Introduction - in-depth analysis of the literature on this organ-sparing multi-modal approach in rectal cancer, 

row 48 - dMMR/MSI-H - abbreviation needs to be explained at first usage - Minor

Material and methods - detailed description of the study protocol - No remarks

Results - properly presented and visualised, firmly substantiated the authors` results - No remarks

Author Response

Title - clearly depicting the purpose of the study - No remarks

Thank you.

Abstract - a concise review of the manuscript, emphasizing all the important points.

Thank you for these comments.

Introduction - in-depth analysis of the literature on this organ-sparing multi-modal approach in rectal cancer, row 48 - dMMR/MSI-H - abbreviation needs to be explained at first usage - Minor

Thank you. The abbreviation has been added on Line 48, in the Introduction section (Page 2).

Material and methods - detailed description of the study protocol - No remarks

Thank you.

Results - properly presented and visualised, firmly substantiated the authors` results - No remarks

Thank you very much.

Round 2

Reviewer 2 Report

Comments and Suggestions for Authors

The author answered all my questions. The manuscript has been sufficiently improved to warrant publication in Cancers.